# Online Optimization Method for Nonlinear Model-Predictive Control in Angular Tracking for MEMS Micromirror

**DOI:** 10.3390/mi13111867

**Published:** 2022-10-30

**Authors:** Qingmei Cao, Yonghong Tan

**Affiliations:** 1Institute of Mechanical Engineering, University of Shanghai for Science and Technology, Shanghai 200093, China; 2College of Mechanical and Electronic Engineering, Shanghai Normal University, Shanghai 200234, China

**Keywords:** hysteresis, Hammerstein architecture, predictive control, micromirror, angular tracking

## Abstract

In this brief, a precise angular tracking control strategy using nonlinear predictive optimization control (POC) approach is address. In order to deal with the model uncertainty and noise interference, a online Hammerstein-model-based POC is designed using online estimated parameters and model residual. Above all, a rate-dependent Duhem model is used to describe the nonlinear sub-model of the whole Hammerstein architecture for depicting multi-valued mapping nonlinear characteristic. Then, predictive output of angular deflection is obtained by Diophantine function based on linear submodel. Subsequently, the iterative control value depends on estimated parameters through data-driven is acquired. Later, based on the cost function, the iteratively optimization control quantity is fed back to the electromagnetic driven deflection micromirror (EDDM) system on the basis of Hammerstein architecture. It should be stressed that the control value is determined by real-time update model residual and defined cost function. Moreover, the stability of POC strategy is proposed. In addition, experimental result is proposed to validate the effectiveness of the control technique adopted in this paper.

## 1. Introduction

Micro Electro-Mechanical System (MEMS) scanning micromirror is the key microchip in Micro-Optic Electro-Mechanical System (MOEMS). As a product of interdiscipline between MEMS and micro-optics, its application field covers a wide range in MEMS field. In fact, electromagnetic driven deflection micromirror (EDDM) is a fast dynamic controllable system with approximate flow type, which has complex dynamic characteristics e.g., nonlinear hysteresis, system oscillation, poor anti-interference, time-varying system structure and high-precision positioning [1,2]. Therefore, the design of controller based on the above characteristics to achieve fast and accurate deflection angle positioning has become its core scientific problem.

In application, the precise positioning control is directly related to the transient and steady state performance of the system. Considering the difficulties in the optimization control of MEMS micromirror system, lots of research institutions have adopted a variety of control strategies to further study. Which includes open-loop control and closed-loop control basically. The former is prone to problems such as long adjustment time and overshoot, which cannot meet the high-performance requirements of the micromirror system [2,3,4]. Hence, most of control strategies for scanning micromirrors belong to closed-loop control.

Most closed-loop control tracking strategies with torsional micromirrors are model-based control methods, e.g., sliding mode control (SMC) [5,6,7], robust control [8,9,10,11], internal model control [7,8], LQG control [12,13], predictive control [14], output regulation control (ORC) [1]. It should be noted that most of the above methods are based on linear or linearized models, which require nonlinear approximation near specific operating points of deflection micromirrors [8,9,10].

In order to avoid the obstacle of modeling for complex nonlinear system, the model independent control strategies e.g., compensated positioning control (OPC) [15] and model-free control (MFC) [16,17] are also adopted to further study. It should be noted that the performance of hysteresis involved in the EDDM could encounter the problem of the nonlinear gradient does not exist. Then, the control effect including transient and steady state is further weakened.

As one of the earliest developed control strategies, PID is widely used in the angle tracking control of scanning micromirrors because of its simplicity, high reliability and no dependence on the system model [18]. The closed-loop PD control scheme is adopted in [19] to solve pull-in for electrostatic micromirror. Whereas, the EDDM is a system with nonlinear hysteresis, the characteristic of multivalued mapping can directly lead to the generation of multiple local working points. The traditional PID parameter setting method is based on a specific operating point, which can not meet the working mode of all operating points of the system. So, the conventional PID with invariable parameters is unable to adapt to time-varying system capabilities. Then, the improved PID controllers are designed to be integrated with sliding mode method [3].

For solving jitter phenomenon in sliding control strategy, robust backstepping sliding mode control is used to improve the transient and stable performance in [20]. Refs. [6,7] contribute the integral sliding mode surface torsion algorithm for closed-loop control of micromirrors by compensating the tracking errors. The improved sliding mode control strategy forces the tracking error to converge to zero along the sliding mode surface, which can suppress the undesirable interference of parameter uncertainty to a certain extent [5]. However, the sliding mode surface only forces the steady-state error to decrease during the switching process, and the chattering phenomenon will not completely disappear. Moreover, this control method does not have the self-adjustment ability to cope with the structural changes of the time-varying system.

In order to obtain the controller adapted to the time-varying system, adaptive control and robust control is the main approaches. Xingling S et al. studied an adaptive quantization controller to achieve disturbance estimation and elimination [21]. A direct adaptive algorithm based on Lyapunov was introduced to find the adaptive law to update the position parameters of the system and estimate the upper bound of the model uncertainty [22]. Although the conservative control effective can be solved by using parameter adaptive to some extent, yet the computational burden caused by online updating cannot be ignored.

Robust control that does not depend on exact model can be used as another option of precise positioning controller design for time-varying systems. Jiazheng T et al. has studied the linear feedback composite integral SMC technique and the robust control method of LQG combined with internal mode to improve the transient and steady-state performance of deflected micromirrors, successively [12]. Chen H et al. adopted a robust nonlinear control method for the two-dimensional laser scanning micromirror system to achieve accurate positioning and tracking of triangular signals [5,9]. However, it needs to meet the requirement of stability margin, which leads to conservative control performance. Moreover, the control characteristics of large uncertainty range lead to poor steady-state accuracy of the controller without optimization process.

Until now, most of control strategies aforementioned are based on the linear or linearization models. This simplified approach can make the controller design of complex nonlinear systems ignore the nonlinear factors and unmodeled disturbances embedded in the system.

In this brief, an online POC based on Hammerstein architecture is proposed for the complex scanning micromirror. To deal with the influence of nonlinear hysteresis, the rate-dependent Duhem model is adopted to depicted the characteristic of EDDM. Assuming the cost function has at least one extreme point, a controller on account of estimated parameters is designed. It should be stressed that the parameters in the recursive expression of the control quantity are estimated in the random noise environment to improve the accuracy of the model.

Compared with the exciting mainstream algorithms, the proposed Hammerstein-based POC has the following advantages. An accurate nonlinear model i.e., Hammerstein model structure is established in random noise environment for describing the complex nonlinear system with short time parameters convergence. Moreover, nonlinear predictive output is obtained using the unmeasured variable, i.e., the deflection torque of the system. In addition, optimization control is designed on the basis of combination of estimated parameters and predictive output at future moments. More than that, the iterative optimization mechanism is also applied to obtain the global optimal value at each sampling time. The above advantages all increase the robustness of the system to achieve high precision angle positioning in noisy environment.

And the control value is iterated in the process of online optimization, which can adjust itself according to the real-time input-output in random system.

The main contributions of this article are as follows:The Hammerstein model for depicting the complex EDDM is established using online input-output data in random noise environment. Moreover, the rate-dependent Duhem submodel is used to describe the hysteresis phenomenon that caused by M−H loop.The mathematical description of control variable can be described using estimated parameters which including linear and hysteresis submodel. In addition, model residual and unmeasurable driven torque, which needs to be estimated is also used to achieve the accurate angular positioning.The polynomial obtained by Diophantine equation can not only be used to describe the prediction equation, but also can directly obtain the influence between deflection angle and driven torque.The residual calculated by the real-time input-output signals and the one-step prediction model are iterated continuously in each sampling time according to the performance index to realize the high-precision positioning of the micromirror system.Rigorous theoretical analysis using discrete Lyapunov candidate function and hardware validation based on real-time signal acquisition for the proposed POC are implemented.

The context and structure of the paper are as follows: in Section 2, the Hammerstein model with small model error is confirmed. Subsequently, the online POC for nonlinear EDDM is proposed in Section 3. Then, Section 4 shows stability analysis using Lyapunov for discrete system. Afterwards, the online physical experiment with comparations are carried out to validate the effectiveness of the proposed scheme in Section 5. Finally, conclusions are analyzed directed at the research content.

## 2. Dynamic Property and Model Establishment

### 2.1. Basic Dynamic Performance Analysis

The proposed EDDM is a typical biaxial scanning micromirror, and the physical design structure is shown in Figure 1. When the two axises i.e., fast and slow scan flexure, are driven by a synthetic signal composed of low frequency and high frequency components, the micromirror can be driven to realize the biaxial deflection.

Actually, the fast axis in static drive is uncontrollable, which is often processed by phase locking to realize the automatic tracking of the output signal frequency to the input signal frequency. Precisely because this processing method inevitably leads to the research object that is defined as a single input single output (SISO) system, which is conducive to subsequent in-depth analysis and research of the system.

To achieve fast scanning, the proposed micromirror is designed as a current driven device. Thus, a complementary push-pull power amplifier circuit with the purpose of converting voltage to current is used to drive the device. From the analysis of input-output data, the fact that the EDDM system exists serious nonlinear hysteretic dynamic performance, which makes the research of precise positioning becomes one of the difficulties in this field. The main reason of nonlinear hysteresis phenomenon is M-H loop caused by electromagnetic effects produced by electromagnetic fields. And the corresponding nonlinear hysteresis phenomenon is shown as Figure 2.

In addition, the oscillation phenomenon caused by the small damping coefficient and the large spring coefficient of the flexural mechanism is particularly obvious [2]. More than that, the electromagnetic micromirror is also affected by process noise and measurement noise [23,24,25].

### 2.2. Architecture of Hammerstein Describing for EDDM

In order to describe the performance of hysteresis and oscillation inherent in the EDDM, a nonlinear model structure with rate-dependent is needed. From the perspective of mechanical structure, it can be seen that the micromirror device can be divided into two parts. One is the driving operation mechanism, which can be described as nonlinear submodel N(·), and the other one is the deflection mechanism, which can be expressed as linear submodel L(·). Using the Hammerstein structure N(·)+L(·) to describe the whole EDDM experimental facility, and it is shown as Figure 3.

From Figure 2, it is obvious that the nonlinear hysteresis has rate-dependent feature. Then, a so-called Duhem model is adopted for describing the driving mechanism N(·), while a linear model is used to depict the deflection mechanism L(·). Considering that electromagnetic effect is the main cause of hysteresis, and the dynamic Duhem model can depict electromagnetic behavior using rate-dependent form [26,27], i.e.,
(1)$$v_{k}=v_{k-1}+\beta \left[\omega \mathrm{sgn}\left(u_{k-1}\right)+\tau u_{k-1}-v_{k-1}\right]\left|\Delta u_{k}\right|+\gamma \Delta u_{k}$$

Therefore, Duhem sub-model is used to describe the nonlinear characteristics inherent in Hammerstein model. Where, $v_{k}$ is the driving torque, which is an immeasurable value, represents the input voltage, control increment $\Delta u_{k-1}=u_{k-1}-u_{k-2}$, β,τ,ω,γ are the coefficients to be estimated.

**Remark** **1.**
*Nonlinear submodel (Equation 1) can be described as $v_{k}=N\left(u_{k-1}\right)\in C^{1}$. Where N(·) is the Lipsitz continuous function with characteristics of piecewise smooth and extreme value.*


A linear submodel with second-order is employed to describe the deflection mechanism. Noted that, the order of the linear model can be confirmed by bode curve diagram through input-output data. After analysis the operation principle, the deflection mechanism can be depicted as:(2)$$J\frac{d^{2}\theta }{dt^{2}}+B\frac{d\theta }{dt}+K\theta =v$$
where, $\theta {(}^{\circ })$ is the angular deflection of the micromirror, v (N·m) is the electromagnetic torque, which is formulated as (Equation 1), $J\;(\mathrm{kg}\cdot m^{2})$ is the moment of inertia, determined by the type and shape of micromirror, B (N·s/m) is the damping coefficient, K (N·m/rad) is the angular spring coefficient of the deflection mechanism; and t (s) is the response time.

By applying the *z*-transformation to (Equation 2), the corresponding linear discrete-time submodel for describing deflection mechanism can be denoted as
(3)$$\theta_{k}=-a_{1}\theta_{k-1}-a_{2}\theta_{k-2}+bv_{k-1}+\varepsilon_{k}$$
where, $a_{1},a_{2},b$ are the coefficients to be estimated, $\varepsilon_{k}$ is system noise that can be regarded as white noise.

**Remark** **2.**
*It is noted that the electromagnetic driven micromirror is seriously polluted by internal and external noise. Considering that the influence of noise factors has an important impact on system, the online modeling method and control strategy should become the preferred choice.*


Substituting (Equation 1) into (Equation 3), a mathematical description with identification form that satisfies the linear parameterization can be obtained
(4)$$\begin{array}{cc} \theta_{k}= & -{\hat{a}}_{1}\theta_{k-1}-{\hat{a}}_{2}\theta_{k-2}+bv_{k-2}+b\hat{\gamma }\Delta u_{k-1}-b\hat{\beta }v_{k-2}\\ & \left|\Delta u_{k-2}\right|+b\hat{\beta }\left(\hat{\omega }\mathrm{sgn}\left(u_{k-2}\right)+\hat{\tau }u_{k-2}\right)\left|\Delta u_{k-2}\right| \end{array}$$

For adapting to the model uncertainty and noise disturbance, the time-varying weight factor is obtained by updating weighting factor with the soften model estimation error. Then, an online data-driven modeling method with time-varying weight factor is proposed [2]. The procedure of parameter convergence and model error are shown in Figure 4 and Figure 5, respectively.

From the results of Figure 4 and Figure 5, we are able conscious of that the Hammerstein architecture with Duhem submodel can describe the dynamic characteristics accurately. Moreover, Figure 4 shows that the parameters of both linear and nonlinear submodels have fast convergence rate. To verify the generalization of the Hammerstein model, another exciting signal is adopted, and the results shows the proposed model has a small output angle deviation with SME (0.09). More than that, but the advantage of the mathematical model is convenient for controller design.

**Remark** **3.**
*It is noted that the designed input signal is assumed to be bounded and can traverse all operating modes of the system described by the Hammerstein dynamic model (Equation 4).*


## 3. Design Philosophy of Nonlinear Predictive Controller

Under the given model architecture in Section 2, if the model can accurately describe the dynamic behavior of the system within the operating range, the controller can be designed to achieve a more accurate angle positioning. In this following content, an online predictive control based on nonlinear Hammerstein architecture with rate-dependent feature is proposed. Compared to the existing literatures, the most obvious advantage is that the predictive control and rate-dependent hysteresis model are combined together to achieve positioning accuracy. Moreover, for preventing fall into local optimum in the process of optimization, a manual adjustment of learning rate is employed, which can also prevent jitter near the local minimum. In addition, the proposed control strategy is equipped with online optimization capability to search the optimal value. The corresponding control system is as shown in Figure 6.

### 3.1. One-Step Predictive Model for Hammerstein Architecture

Actually, the linear submodel (Equation 3) can be depicted as ARIMA model:(5)$$\hat{A}\left(z^{-1}\right)\theta_{k}=z^{-1}\hat{B}\left(z^{-1}\right)v_{k}+\varepsilon_{k}$$
where,
(6)$$\hat{A}\left(z^{-1}\right)=1+{\hat{a}}_{1}z^{-1}+\ldots +{\hat{a}}_{n_{a}}z^{-n_{a}},\deg A\left(z^{-1}\right)=n_{a}$$
(7)$$\hat{B}\left(z^{-1}\right)={\hat{b}}_{0}+{\hat{b}}_{1}z^{-1}+\ldots +{\hat{b}}_{n_{a}}z^{-n_{b}},\deg B\left(z^{-1}\right)=n_{b}$$

Herein, ${\hat{a}}_{i}$ and ${\hat{b}}_{i}$ are the coefficients of linear model (Equation 3) obtained by data-driving, which can describe the oscillatory properties; $z^{-1}$ is backward shift operator, $n_{a}$ and $n_{b}$ are the orders of the polynomial $\hat{A}\left(z^{-1}\right)$ and $\hat{B}\left(z^{-1}\right)$.

Define the Diophantine equation:(8)$$1=E_{j}\left(z^{-1}\right)\hat{A}\left(z^{-1}\right)\Delta +z^{-j}F_{j}\left(z^{-1}\right)$$
where, $E_{j}\left(z^{-1}\right)$ and $F_{j}\left(z^{-1}\right)$ are polynomial uniquely defined by $\hat{A}\left(z^{-1}\right)$ and prediction length *j*, which can be solved by recursive method. Since one-step prediction is adopted in this present paper, i.e., j=1, the corresponding Diophantine equation can be written as
(9)$$1=E_{1}\left(z^{-1}\right)\tilde{A}\left(z^{-1}\right)+z^{-1}F_{1}\left(z^{-1}\right).$$

Therefore, $E_{1}\left(z^{-1}\right)=1$, $F_{1}\left(z^{-1}\right)=z\left[1-\tilde{A}\left(z^{-1}\right)\right]$, $\tilde{A}\left(z^{-1}\right)=\hat{A}\left(z^{-1}\right)\Delta$. Subsequently, one-step predictive model can be acquired as:(10)$${\hat{\theta }}_{k+1/k}=E_{1}\left(z^{-1}\right)\hat{B}\left(z^{-1}\right)\Delta v_{k/k}+F_{1}\left(z^{-1}\right)\theta_{k}$$

### 3.2. Controller Design Based on Optimization Performance

The original intention of controller design for EDDM is to accurately track a set-trajectory in order to obtain small output angle deviation. Then, the cost function can be defined as
(11)$$Q_{c}\left(e_{k+1},u_{k}\right)=\frac{1}{2}e_{k+1}^{2}+\frac{\kappa }{2}\Delta u_{k}^{2},$$
where, κ is the weighting factor related to control process and control increment, which needs to be satisfy κ>0, and the tracking error is
(12)$$e_{k+1}=\theta_{r_{k}}-{\hat{\theta }}_{k+1/k}$$
where, $\theta_{r_{k}}$ is the set-trajectory at time *k*.

**Assumption** **1.**
*The reference trajectory $\theta_{r_{k}}$ is piecewise smooth and bounded, i.e., $\theta_{r_{k}}\in \Omega_{r}$, where $\Omega_{r}$ is a bounded compact set.*


**Assumption** **2.**
*The cost function $Q_{c}\left(e_{k+1},u_{k}\right)$ has at least one extreme value in defined domain.*


Based on Assumptions 1 and 2, the influence of $u_{k}$ on the tracking error can be obtained using gradient descent algorithm to (Equation 11):(13)$$\frac{\partial Q_{c}\left(e_{k+1},u_{k}\right)}{\partial u_{k}}=\kappa \Delta u_{k}+\frac{\partial e_{k+1}}{\partial u_{k}}e_{k+1}.$$

Based on (Equation 12), define
(14)$$\frac{\partial e_{k+1}}{\partial u_{k}}=-\frac{\partial {\hat{\theta }}_{k+1/k}}{\partial u_{k}}=-\Lambda \left(u_{k-1},v_{k-1}\right).$$

For searching the extreme value, let
$$\frac{\partial Q_{c}\left(e_{k+1},u_{k}\right)}{\partial u_{k}}=0.$$

Subsequently, the optimal control strategy based on Hammerstein structure is designed as
(15)$$u_{k}=u_{k-1}+\lambda \Lambda \left(u_{k-1},v_{k-1}\right)e_{k+1}$$
where, λ=11κκ is the optimizing step-size.

Considering the proposed control algorithm is based on Hammerstein model, and the driving torque in the model structure is an unknown intermediate variable, then based on (Equation 1) and (Equation 10), we will obtain
(16)$$\Lambda \left(u_{k-1},v_{k-1}\right)=\frac{\partial {\hat{\theta }}_{k+1/k}}{\partial v_{k}}\frac{\partial v_{k}}{\partial u_{k}}$$

And the designed controller can be depicted as
(17)$$u_{k}=u_{k-1}+\lambda \frac{\partial {\hat{\theta }}_{k+1/k}}{\partial v_{k}}\frac{\partial v_{k}}{\partial u_{k}}e_{k+1}$$

**Remark** **4.**
*Utilize the solution of Diophantine function of one-step predictive, then one can obtain*

(18)
$$\frac{\partial {\hat{\theta }}_{k+1/k}}{\partial v_{k}}=B\left(z^{-1}\right).$$



In view of (Equation 1),
(19)$$\frac{\partial v_{k}}{\partial u_{k}}=\beta \left[\omega \mathrm{sgn}\left(u_{k-1}\right)+\tau u_{k-1}-v_{k-1}\right]+\gamma .$$

Equation (Equation 14) can be described as
(20)$$\frac{\partial e_{k+1}}{\partial u_{k}}=-\frac{\partial {\hat{\theta }}_{k+1/k}}{\partial u_{k}}=-\frac{\partial {\hat{\theta }}_{k+1/k}}{\partial v_{k}}\frac{\partial v_{k}}{\partial u_{k}}=-\Lambda \left(u_{k-1},v_{k-1}\right).$$

Combine the above three formulas (Equation 18)–(Equation 20), we can confirm that the output function is related to the independent variable $u_{k-1}$ and $v_{k-1}$, where $v_{k-1}={\hat{v}}_{k-1}$ is an estimated value.

Subsequently, using (Equation 17)–(Equation 19), a recursive expression for the optimal controller based on Hammerstein architecture is obtained
(21)$$u_{k}=u_{k-1}+\lambda B\left(z^{-1}\right)\left[\beta \left[\omega \mathrm{sgn}\left(u_{k-1}\right)+\tau u_{k-1}-v_{k-1}\right]+\gamma \right]e_{k+1}.$$

Noted that the parameters including nonlinear submodel β,ω,τ,γ along with linear submodel $B(z^{-1})$ can be approximated to the estimated $\hat{\beta },\hat{\omega },\hat{\tau },\hat{\gamma }$,$\hat{B}\left(z^{-1}\right)$ within a small model error.

In order to search the optimal solution, a corresponding optimization algorithm is carried out in each sampling period. The iterative optimization process is as Figure 7.

## 4. Stability Analysis

A discrete Lyapunov candidate function is chosen to analyze the stability of the Hammerstein architecture-based model with one-step predictive optimal control strategy for EDDM. And the corresponding analysis is as following.

**Theorem** **1.**
*Suppose the control strategy depicted by (Equation 17) is adopted to system (Equation 5). If the weighting factor of Lyapunov function is selected as*

(22)
$$\sigma =\varpi +2\Lambda^{2}\left(u_{k-1},v_{k-1}\right)$$

*where ϖ>0. And if the optimizing step satisfies*

(23)
$$0<\lambda \leq \frac{2}{\varpi +\Lambda^{2}\left(u_{k-1},v_{k-1}\right)}$$


*Then,*

(24)
$$\lim_{k\to \infty }e_{k+1}\to e^{*}$$

*where, $e^{*}$ denotes the minimum angle tracking deviation.*


**Proof.** Choose Lyapunov function
(25)$$V_{k}=e_{k+1}^{2}+\sigma \Delta u_{k}^{2}.$$In order to obtain the relation of tracking error at adjacent moments, the following approximate calculation is made
(26)$$e_{k+1}=e_{k}+\Delta e_{k+1}\approx e_{k}+\frac{\partial e_{k+1}}{\partial u_{k}}\Delta u_{k}.$$Combined with (Equation 14), it yields
(27)$$\Delta e_{k+1}\approx \frac{\partial e_{k+1}}{\partial u_{k}}\Delta u_{k}=-\Lambda \left(u_{k-1},v_{k-1}\right)\Delta u_{k}.$$Based on (Equation 15), there is
(28)$$\Delta u_{k}=\lambda \Lambda \left(u_{k-1},v_{k-1}\right)e_{k+1}.$$Carry out difference calculation according to (Equation 25), one can obtain
(29)$$\begin{array}{cc} V_{k}-V_{k-1} & =e_{k+1}^{2}+\sigma \Delta u_{k}^{2}-e_{k}^{2}-\sigma \Delta u_{k-1}^{2}\\ & ={\left[e_{k}-\Lambda \left(u_{k-1},v_{k-1}\right)\Delta u_{k}\right]}^{2}+\sigma \Delta u_{k}^{2}-e_{k}^{2}-\sigma \Delta u_{k-1}^{2}\\ & =\Lambda^{2}\left(u_{k-1},v_{k-1}\right)\Delta u_{k}^{2}-2\Lambda \left(u_{k-1},v_{k-1}\right)\Delta u_{k}e_{k}+\sigma \Delta u_{k}^{2}-\sigma \Delta u_{k-1}^{2}. \end{array}$$In view of (Equation 28), (Equation 29) can be described as
(30)$$\begin{array}{cc} V_{k}-V_{k-1} & =\lambda^{2}\Lambda^{4}\left(u_{k-1},v_{k-1}\right)e_{k+1}^{2}-2\lambda \Lambda^{2}\left(u_{k-1},v_{k-1}\right)e_{k+1}e_{k}\\ & +\sigma \lambda^{2}\Lambda^{2}\left(u_{k-1},v_{k-1}\right)e_{k+1}^{2}-\sigma \Delta u_{k-1}^{2}\\ & \leq \lambda \Lambda^{2}\left(u_{k-1},v_{k-1}\right)\left[\hbar e_{k+1}^{2}\left(\Lambda^{2}\left(u_{k-1},v_{k-1}\right)+\sigma \right)-2e_{k+1}e_{k}\right]. \end{array}$$Apparently, if
(31)$$\lambda e_{k+1}^{2}\left(\Lambda^{2}\left(u_{k-1},v_{k-1}\right)+\sigma \right)-2e_{k+1}e_{k}\leq 0$$
i.e.,
(32)$$0<\lambda \leq \frac{2e_{k+1}e_{k}}{\left(\Lambda^{2}\left(u_{k-1},v_{k-1}\right)+\sigma \right)e_{k+1}^{2}}.$$Then, one can obtain $V_{k}\leq V_{k-1}$. Based on (Equation 26) and (Equation 28)
(33)$$e_{k+1}=e_{k}-\lambda \Lambda^{2}\left(u_{k-1},v_{k-1}\right)e_{k+1}.$$Multiply both sizes by $e_{k+1}$, then
(34)$${e^{2}}_{k+1}=e_{k+1}e_{k}-\lambda \Lambda^{2}\left(u_{k-1},v_{k-1}\right){e^{2}}_{k+1}.$$Substituting (Equation 34) into (Equation 32), then
(35)$$0<\lambda \leq \frac{2(1+\hbar \Lambda^{2}\left(u_{k-1},v_{k-1}\right))}{\Lambda^{2}\left(u_{k-1},v_{k-1}\right)+\sigma }.$$After rearranging the above equation, we will have
(36)$$0<\lambda \leq \frac{2}{\sigma -\Lambda^{2}\left(u_{k-1},v_{k-1}\right)}.$$In combination with (Equation 22), it yields
(37)$$0<\lambda \leq \frac{2}{\varpi +\Lambda^{2}\left(u_{k-1},v_{k-1}\right)}.$$The constraints of optimization step size leads to
(38)$$V_{k}-V_{k-1}\leq 0.$$That implies the energy function $V_{k}$ is a decreasing sequence.Suppose $e^{*}=\min\left\{e_{k+1}\right\}$ exist, under the definition of
(39)$${\tilde{e}}_{k+1}=e_{k+1}-e^{*},$$
subtract $e^{*}$ from both sides of (Equation 33), subsequently
(40)$${\tilde{e}}_{k+1}={\tilde{e}}_{k}-\lambda \Lambda^{2}\left(u_{k-1},v_{k-1}\right)e_{k+1}.$$Based on (Equation 40)
(41)$$\begin{array}{c} {\tilde{e}}_{k+1}^{2}={\tilde{e}}_{k}^{2}-2\lambda \Lambda^{2}\left(u_{k-1},v_{k-1}\right)e_{k+1}^{2}\left(1-e^{*}/e_{k+1}\right)-\lambda^{2}\Lambda^{4}\left(u_{k-1},v_{k-1}\right)e_{k+1}^{2} \end{array}.$$For λ=1/κ>0, and
(42)$$\left\{\begin{array}{cc} 1-e^{*}/e_{k+1}=0 & e^{*}=e_{k+1}\\ 0<1-e^{*}/e_{k+1}<2 & otherwise \end{array}\right.$$
both with
(43)$${\tilde{e}}_{k+1}^{2}\leq {\tilde{e}}_{k}^{2}.$$Therefore, $\left\{{\tilde{e}}_{k+1}^{2}\right\}$ is defined as a decreasing sequence, i.e., ${\tilde{e}}_{k+1}^{2}\leq {\tilde{e}}_{k}^{2}\leq \ldots \leq {\tilde{e}}_{0}^{2}$. It implies there exists a nonnegative scalar ${\left({\tilde{e}}^{*}\right)}^{2}$ satisfies $\lim_{k\to \infty }{\tilde{e}}_{k+1}^{2}={\left({\tilde{e}}^{*}\right)}^{2}$.Using (Equation 40), we will have
(44)$$\frac{{\tilde{e}}_{k+1}^{2}}{{\tilde{e}}_{k}^{2}}=\frac{1}{{[1+\lambda \Lambda^{2}\left(u_{k-1},v_{k-1}\right)/\left(1-e^{*}/e_{k+1}\right)]}^{2}}.$$Define
(45)$$\varphi_{k}==1/[1+\lambda \Lambda^{2}\left(u_{k-1},v_{k-1}\right)/\left(1-e^{*}/e_{k+1}\right){]}^{2}.$$Based on (Equation 41) and (Equation 44), under the definition of $\varphi =\max\{\varphi_{k}\}$,
(46)$$0\leq {\tilde{e}}_{k+1}^{2}\leq \varphi {\tilde{e}}_{k}^{2}\leq \varphi_{}^{2}{\tilde{e}}_{k-1}^{2}\leq \ldots \leq \varphi_{}^{k+1}{\tilde{e}}_{0}^{2}$$After analyzing (Equation 44), there are two cases need to discuss.Case A: $\varphi_{k}=1$In this case, $\Lambda (u_{k-1},v_{k-1})=0$. Based on (Equation 39) and (Equation 44), it means ∀ξ>0 satisfies $\left|e_{k+1}-e^{*}\right|<\xi$, i.e., $\lim_{k\to \infty }e_{k+1}=e^{*}$.Case B: $\varphi_{k}=1$$\Lambda (u_{k-1},v_{k-1})\neq 0$, under this circumstance, if $e^{*}\neq e_{k+1}$, based on (Equation 46) and ${\tilde{e}}_{0}^{2}\in L_{\infty }$, we can have
(47)$$\lim_{k\to \infty }e_{k+1}^{2}=0.$$Therefore, according to above two cases, it leads to
$$\lim_{k\to \infty }e_{k+1}=e^{*}.$$ □

## 5. Experimental Results

In order to evaluate the performance of the proposed POC, conducted quantitative experiments are implemented on the platform of electromagnetic driven deflection micromirror. The corresponding experimental equipment is shown in Figure 8. In the control process, the deflection angle is measured by the piezoresistive sensor based on the chip of EDDM. In fact, the measured voltage signal which is approximately linear to the deflection angle is very small, so a push-pull amplifier circuit is used to amplify the output voltage signal. Subsequently, the amplified signal is fed to the PC using A/D converter (DS2004-BL1), which based on the real-time signal acquisition platform. Meanwhile, this amplified signal is also used to generate the optimizing control value via the cost function and set-trajectory [2,14]. The corresponding control framework is shown in Figure 9, which including an electromagnetic scanning micromirror, a driving circuit, a dSpace real-time acquisition platform and a PC.

**Remark** **5.**
*As the EDDM is essentially a current-driven device, it needs a voltage-current conversion circuit to complete the driving operation. Moreover, it is necessary to pay attention to the driving current which needs be limited within 100 mA during the experiment process. Thus, hardware and software current limiting devices such as saturation functions and current limiting protection circuits are applied to both algorithms and hardware devices to protect EDDM integrated chips.*


### 5.1. Effect of Optimization Step-Size on Dynamic Performance

#### 5.1.1. Effect of Step-Size λ

Parameter λ is optimizing step-size which can directly affect the control performance of the system. In order to analyze the effect of optimizing step-size on the control performance of EDDM, let λ be 0.2,0.33 and 0.4, respectively. Figure 10 shows the response of the proposed control strategy when λ is chosen as different values. From which, we can see clearly that λ=0.33 results in the most steady-state performance. The selection of optimizing step-size λ affects the positioning accuracy of the scanning micromirror directly. According to Equation (Equation 21), it is not difficult to understand that there is a time-varying proportional mapping relationship between the input control quantity and the residual of the system model residual $e_{k}+1$.

#### 5.1.2. Comparation with PID Excited by Square Wave

To evaluate the effectiveness of the algorithm, the PID experiment with P=0.95,
$I=0.08,D=1\times 10^{-10}$ is used for comparation. Noted that, because EDDM has fast dynamic characteristics, the commonly used PID parameter tuning methods such as critical proportion method and decay curve method are not suitable for this system, so the trial and error method is used to adjust the parameters of the controlled system in the process of the experiment to obtain good dynamic performance. For reducing positioning errors, the optimizing step-size λ is selected as λ=0.33. The corresponding experimental results are shown as Figure 11 and Figure 12, where the set-trajectory is set as a square wave signal with varying amplitude.

The experiment of deflection angle via proposed POC and PID are shown as Figure 11 and Figure 12. From the comparation results between the two methods, the overshoot of the proposed POC control is small, and even no overshoot can be achieved on the step response of time 1.7 s during the process of increasing the deflection angle. As for the process of decreasing deflection angle, the steady-state performance of the proposed POC method is significantly better than that of the PID control method.

In addition, mean square error (MSE) is also analyzed in Table 1, it can be seen clearly that the PID method generates lager tracking error (MSE =0.0628) than that produced by proposed method (MSE =0.0404). After in-depth analysis of this phenomenon, the reasons is as follows: ➀ Direct use of the error signal causes the control moment to exhibit inertial characteristics; ➁ No ability to seek optimal control through cost function. The above explanations can well explain that the control voltage in Figure 12 is not as stable as in Figure 11. Moreover, considering the transient process, the proposed strategy has shorter adjustment time (0.01 s) with smaller overshoot (7.29%) compared with PID that processes adjustment time (0.05 s) with overshoot (12.66%), which are depicted in the enlarged graphics in Figure 11 and Figure 12.

### 5.2. Result of Experiment Based on POC

In order to evaluate the system response in the environment where the amplitudes and frequencies are constantly changing instead of that where the amplitudes change abruptly, a sinusoidal signal is adopted to excite the EDDM system. The corresponding real-time deflection angle output with the control value which fed back to the system using the proposed POC strategy is as shown in Figure 13. Noted that the optimizing step-size is selected as λ=0.3225 and the sampling period *T* = 0.0001 s.

For comparing with the proposed method, a PID control scheme is used to the system control of EDDM. The control respond is shown in Figure 14 with the corresponding experimental parameters P=0.95,I=0.01,D=0.0005. Due to the approximation as a flow-type system, trial tuning method is adopted for PID parameters confirmation.

In addition, the IMC strategy is also implemented to the EDDM system for comparation. As a model-based robust control scheme, the IMC method is usually adopted in the system which affected by environment seriously. It should be stressed that, in order to acquire the designed controllor of IMC, a first order filter is adopted. Moreover, the IMC controller composed of minimum phase of the EDDM mode and filter is developed. In the process of model inversion, the filter with minimum gain should be selected to ensure the stabilization of the control system. Figure 15 illustrates the control response of the IMC strategy.

To further evaluate the control performance of the three control strategies in detail, Figure 16 illustrates the partial enlarged diagram from 0.1∼0.163 s of the control results using POC, PID and IMC, respectively. And the corresponding control errors of comparation are shown in Figure 17. It is seen clearly that the proposed POC control strategy has the best control effect with the smallest control error of $0.181^{\circ }$. Even when achieving small deflection angle, the tracking error is small, but the positioning and tracking effect of large deflection angle is not obvious. After analysis from Figure 16 and Figure 17, the PID method has smaller deflection angle error $0.223^{\circ }$ compared with IMC scheme $0.7^{\circ }$.

The reason for this phenomenon is that the nonlinear is more serious when the deflection angle achieves a large deflection angle, the PID control algorithm doesn’t have the ability to follow quickly, thus, the phase lag is generated at the same time.

## 6. Conclusions

In order to achieve precise angular deflection of slow axis in EDDM, the POC scheme based on a so-called Hammerstein model is proposed. Considering the complex performance of EDDM system, the Hammerstein architecture is used to describe the whole system, in which the nonlinear and linear submodels are used to describe the driving mechanism and deflection mechanism, respectively.

A predictive control strategy for nonlinear system is proposed. The description of predictive angular deflection contains output of nonlinear submodle, i.e., driving torque, which is unmeasurable value. Then, the optimal control strategy based on Hammerstein architecture is acquired using gradient descent algorithm. Actually, the iterative expression of the designed controller uses both linear and hysteresis submodel, which needs to be estimated in random noise system. The experimental results for EDDM show that the proposed nonlinear POC is able to fast and accurate angular positioning with different track set-trajectories.

The Hammerstein-model-based POC scheme is developed on the premise that the exact nonlinear model is known. This not only requires the model to meet the requirements of high accuracy and generalization, but also puts forward high requirements on the convergence speed, convergence effect and dissipation of parameters. Moreover, long research period to acquire exact linear model is another obstacle for engineering application. Thus, how to use simplified model to design controller to obtain satisfactory dynamic performance is the focus of the further research.

## Figures and Tables

**Figure 1 micromachines-13-01867-f001:**
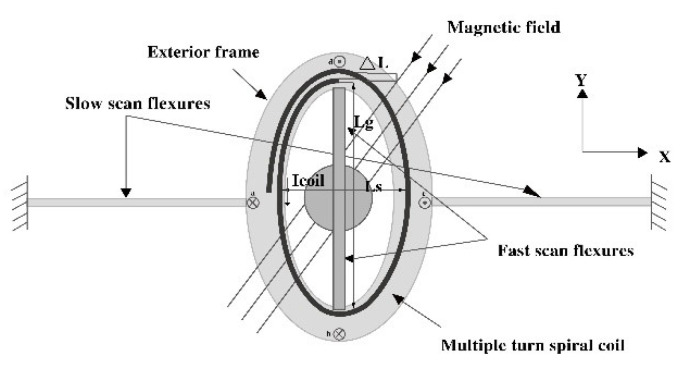
Schematic view of EDDM structure.

**Figure 2 micromachines-13-01867-f002:**
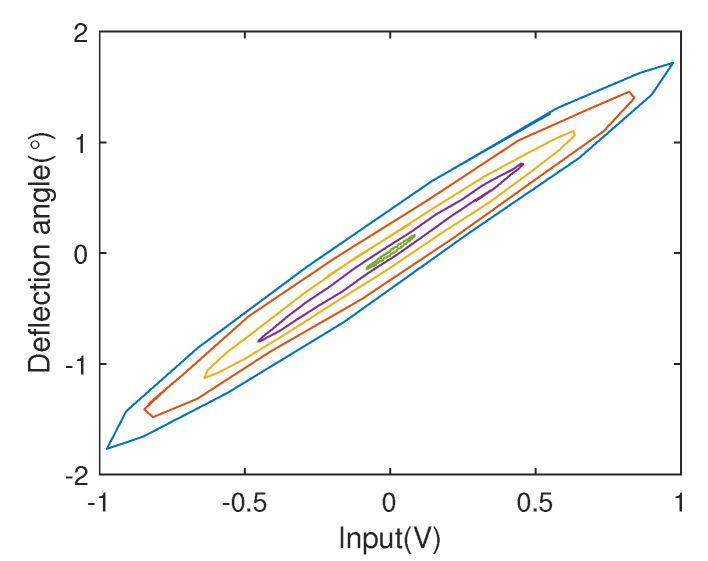
Nonlinear hysteresis phenomenon in EDDM.

**Figure 3 micromachines-13-01867-f003:**
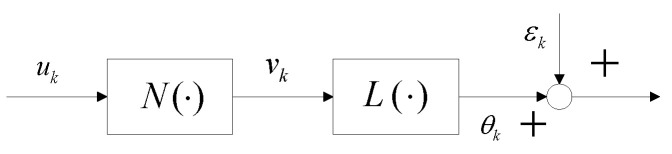
Hammerstein structure for the EDDM.

**Figure 4 micromachines-13-01867-f004:**
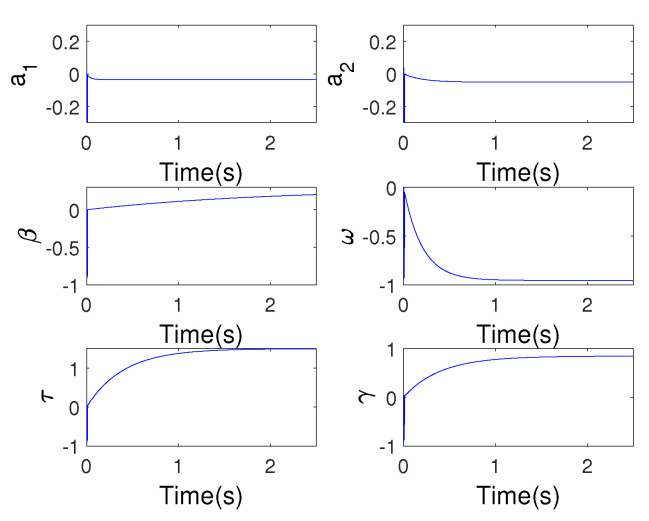
Convergence procedure of estimated parameters for Hammerstein model.

**Figure 5 micromachines-13-01867-f005:**
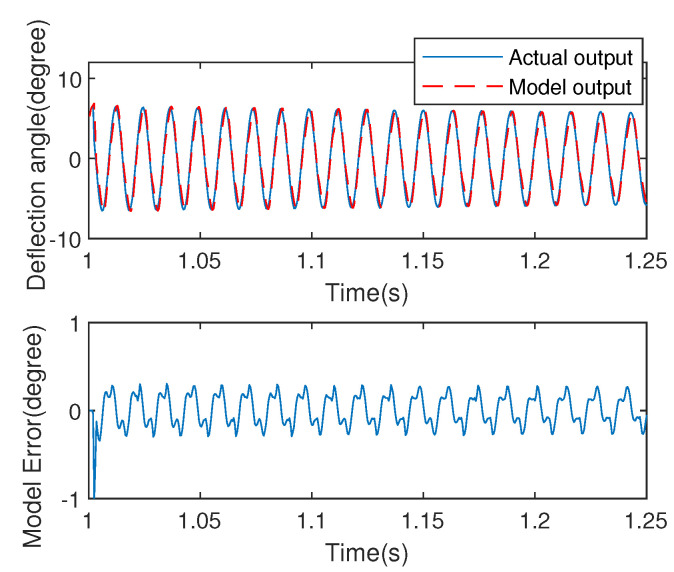
Model validation for Hammerstein model.

**Figure 6 micromachines-13-01867-f006:**
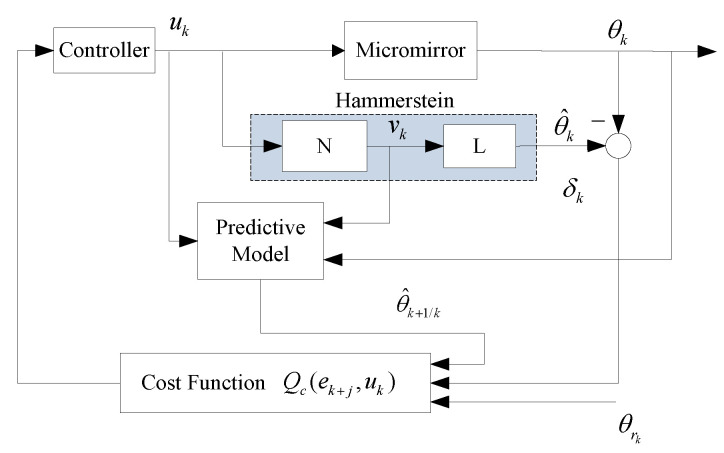
Control framework with predictive optimization based on Hammerstein structure.

**Figure 7 micromachines-13-01867-f007:**
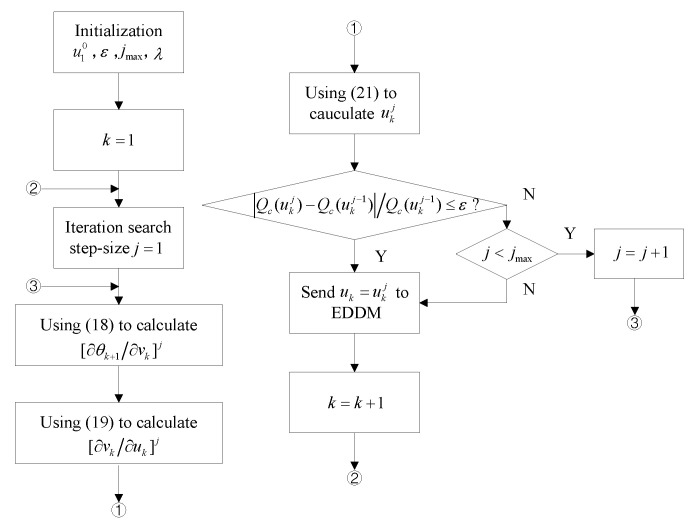
Flowchart of POC.

**Figure 8 micromachines-13-01867-f008:**
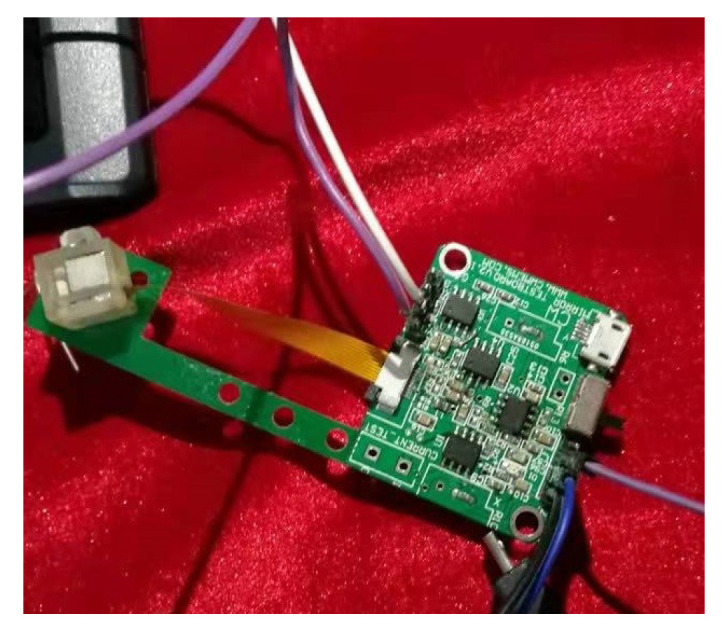
Experimental equipment.

**Figure 9 micromachines-13-01867-f009:**
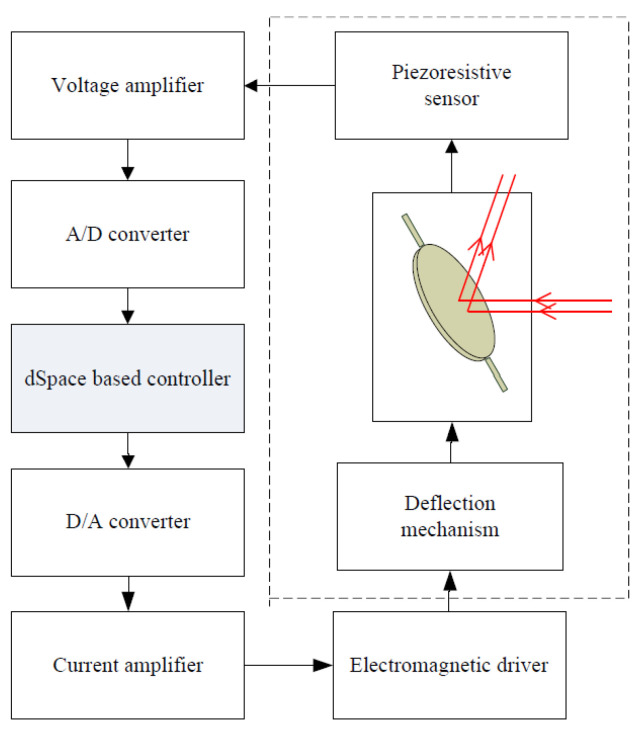
Diagram for EDDM based angular positioning.

**Figure 10 micromachines-13-01867-f010:**
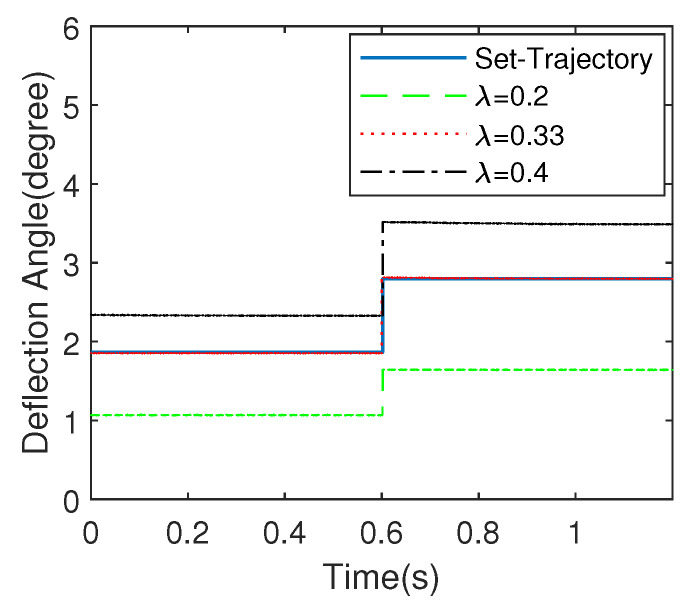
Effect of step-size λ to the EDDM.

**Figure 11 micromachines-13-01867-f011:**
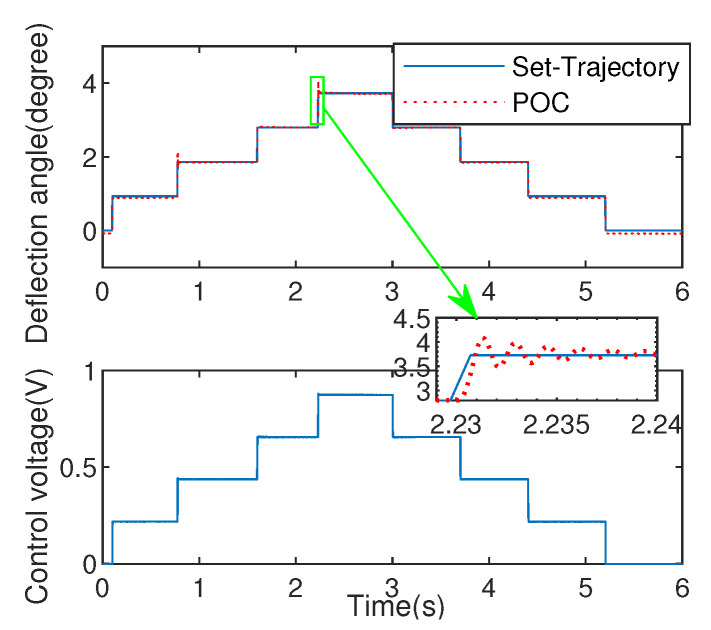
Control effect via POC.

**Figure 12 micromachines-13-01867-f012:**
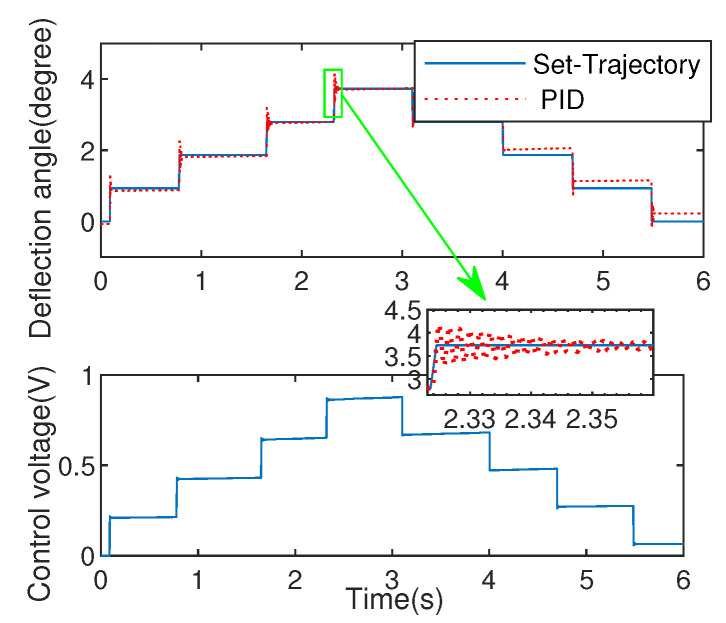
Control effect via PID.

**Figure 13 micromachines-13-01867-f013:**
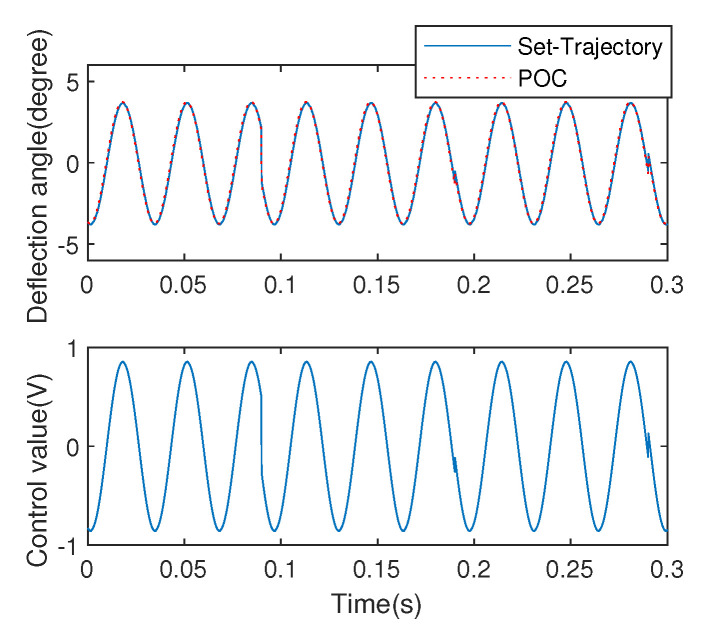
Angular positioning control using POC.

**Figure 14 micromachines-13-01867-f014:**
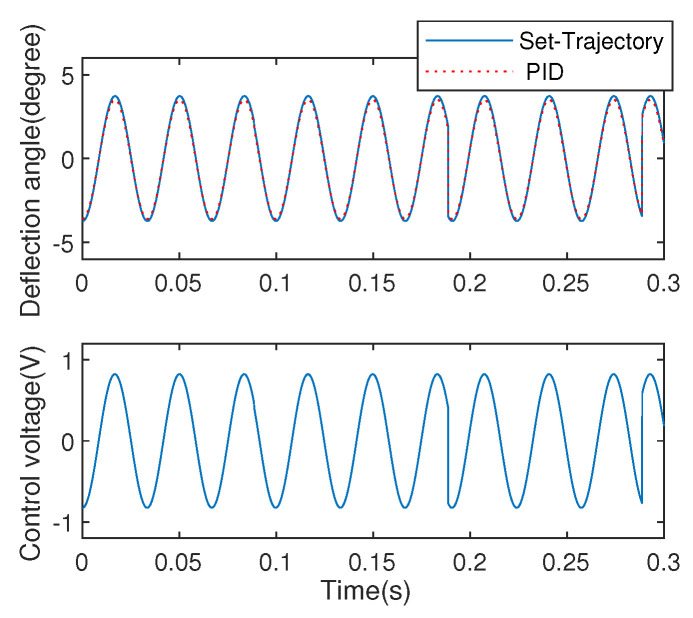
Angular positioning control using PID.

**Figure 15 micromachines-13-01867-f015:**
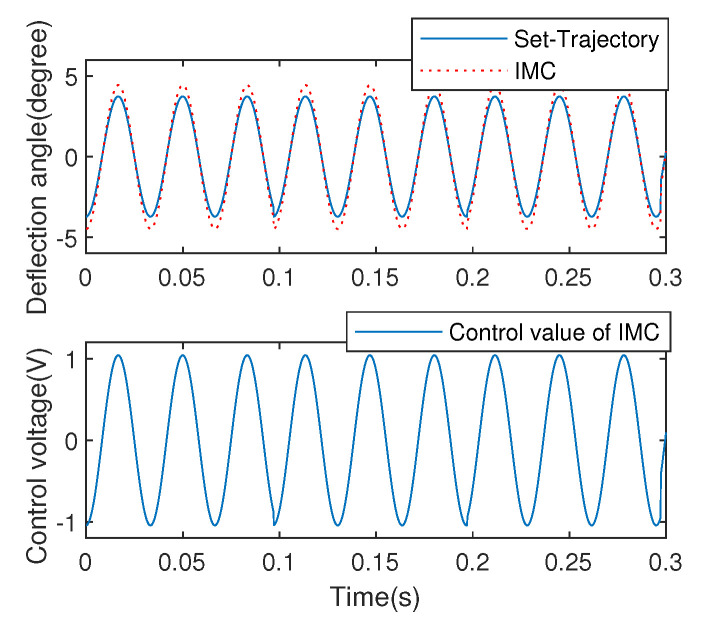
Angular positioning control using IMC.

**Figure 16 micromachines-13-01867-f016:**
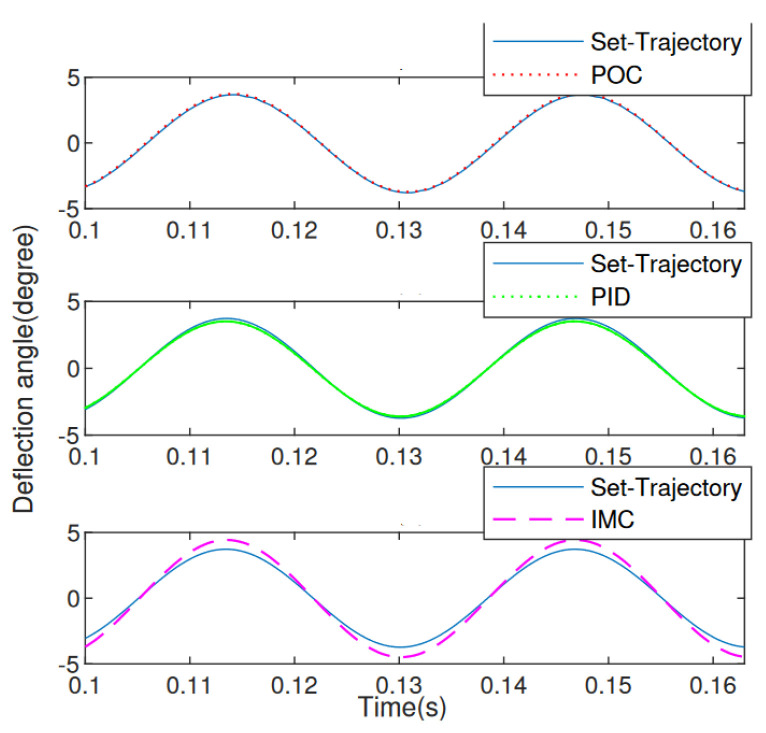
Comparation of control response of POC, PID and IMC via enlarged view.

**Figure 17 micromachines-13-01867-f017:**
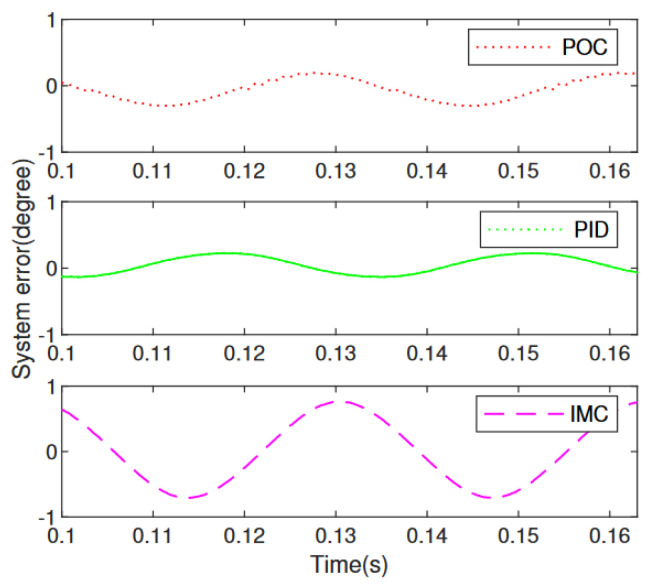
Comparation of system error of POC, PID and IMC.

**Table 1 micromachines-13-01867-t001:** Comparation of performance index for control methods.

Control Method	MSE
POC	0.0404
PID	0.0628

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
