# Peer review of "Online Optimization Method for Nonlinear Model-Predictive Control in Angular Tracking for MEMS Micromirror"

_micromachines, 2022, doi:10.3390/mi13111867_

Round 1

Reviewer 1 Report

This manuscript presents the nonlinear predictive optimization control (POC) scheme based on Hammerstein model to achieve precise angular deflection of slow axis in an electromagnetic driven deflection micromirror (EDDM). Experimental results for EDDm indicated that the proposed nonlinear POC can be used for angular positioning with different track set-trajectories. This manuscript can be improved with the following comments:
1.-The abstract should include a conclusion.
2.-Which are the advantages and limitations of the proposed POC scheme?
3.-Label "defelction" must be corrected in the figures 14, 15 and 16.
4.-The description of the experimental equipment should include more technical information of the main characteristics or references.
5.-The discussions of the results of figures 12-16 should be enhanced.

Author Response

Dear Reviewers,

    Thanks for your comments. Please see the attachment.

Sincerely,

The authors

Reviewer 2 Report

This paper presents a solution to the angular positioning problem for MEME micromirror using nonlinear predictive optimization control. Specifically, a Hammerstein structure is used to describe the electromagnetic driven deflection micromirror (EDDM) with rate-dependent Duhem submodel embedded in it. Then, Hammerstein model-based predictive optimization control (POC) strategy is developed to achieve accuracy angular positioning. The online experiment provided by the authors for the designed algorithm shows that their algorithm can successfully control the nonlinear system with low tracking error. In the reviewer's opinion, the research object of this article is novel and has good application prospects and scientific research value. However, in order to further improve the quality of the paper, the following comments should be considered in the revision.

1
What are the main motivation of the authors for the presentation of an predictive  optimization controller (POC) for EDDM from a practical viewpoint?

2Compared to the existing predictive control results for MEMS micromirror, such as [14] and the control results of adaptive control [21], what are the advantages of the method proposed in this paper? 

3More explanation on the data-driven method for parameter convergence and model error of Figure 4 and Figure 5.

4On page 7/15, right column, lines 104 through 105 authors discusseConsidering the proposed control algorithm is based on Hammerstein model, and the model structure is an unknown intermediate variable…” .This sentence is not clear and should be modified.

5Overall, I have found some typos and grammatical errors while reading the paper, for instance. The authors are suggested to proofread the whole paper to improve the readability.

Author Response

(The authors gave the same response as above.)

Round 2

Reviewer 1 Report

The authors have improved their manuscript considering the reviewer's comments.